# Early Diagnosis of Pine Wilt Disease in *Pinus thunbergii* Based on Chlorophyll Fluorescence Parameters

**Fei Liu** [1,2]**, Mao-Jiao Zhang** [1,2]**, Jia-Feng Hu** [1,2]**, Min Pan** [1,2]**, Lu-Yang Shen** [1,2]**, Jian-Ren Ye** [1,2] **and Jia-Jin Tan** [1,2]*****

[1] Co-Innovation Center for Sustainable Forestry in Southern China, College of Forestry, Nanjing Forestry University, Nanjing 210037, China

[2] Jiangsu Key Laboratory for Prevention and Management of Invasive Species, Nanjing 210037, China

* Correspondence: tanjiajin@njfu.edu.cn; Tel.: +86-137-7070-6308

**Abstract:** As the most severe forestry quarantine disease in several countries, pine wilt disease (PWD) causes substantial economic losses and poses a significant threat to the forest ecosystem. It is necessary to find a rapid and sensitive method for the early diagnosis of the disease to control the development of the disease effectively. This paper investigated the effect of *Bursaphelenchus xylophilus*（the pinewood nematode; PWN）on the chlorophyll fluorescence kinetic curve (OJIP curve) and the parameters of needles using four-year-old *Pinus thunbergii* as experimental materials and chlorophyll fluorescence analysis as a technical tool. It was shown by the results in the OJIP curve that the fluorescence intensity of the inoculated plants was significantly increased at points O and I. Additionally, the relative variable fluorescence intensity at points K and J was comparable to that of the control plants. Several chlorophyll fluorescence parameters of the treatment significantly increased or decreased with disease progression. At the same time, the control group had no significant changes in each parameter. Therefore, chlorophyll fluorescence parameters can be used as indicators for the early diagnosis of PWD, among which the DIo/RC parameter was the best. In summary, PWN invasion will produce fluorescence on the PSII of *P. thunbergii*, and its chlorophyll fluorescence parameters are expected to achieve early PWD diagnosis.

**Keywords:** *Bursaphelenchus xylophilus*; chlorophyll fluorescence kinetic curve; chlorophyll fluorescence parameters; early diagnosis; *Pinus thunbergii*

## 1. Introduction

Pine trees are widely distributed worldwide and are mainly concentrated in northern hemisphere countries. These trees have essential economic and ornamental values. Currently, the pinewood nematode (PWN), *Bursaphelenchus xylophilus*, is thought to be transmitted from North America [1], and the pine wilt disease (PWD) it causes is a devastating pine tree disease [2]. PWD has caused catastrophic damage to pine forest resources in Japan [2] and poses a severe threat to pine forest resources in China, Korea, Portugal, and Spain [3–7]. The pathogenic mechanism of PWN has not been clarified, thus leading to unsolved PWD control challenges. The current disease early diagnosis technology is limited by technical means and conditions to meet disease prevention and control [8], and further research is needed.

Pine trees usually die within a brief period after being infected with PWN. In the early stages of disease development, many defense substances, such as terpenoids and reactive oxygen species (ROS), are produced in pines to resist or inhibit the invasion of PWN. Terpenoids are subdivided into monoterpenoids, sesquiterpenoids, and polyterpenoids, with α-pinene being the most common monoterpenoid secreted by conifers [9]. Hydrogen peroxide (H2O2) is the most stable ROS in pine [10]. Along with the aggravation of the disease, the excessive accumulation of hydrogen peroxide in the tree leads to damage to the pine tree itself. $H_2O_2$ can freely cross the cell membrane, leading to the

peroxidation of internal cellular components, weakening organelle function, and potentially causing cell death [11]. The rapid increase in the content of monoterpenes and sesquiterpenes in the pine tree triggers the production of air bubbles in tubular xylem cells, causing blockage of water transport and eventually leading to death [12]. At the middle and late stages of disease development, the net photosynthetic rate, stomatal conductance [13], and chlorophyll content decreased significantly [14]. This indicates that PWD may cause damage to the photosynthetic machinery of pine needles.

The chlorophyll fluorescence technique uses chlorophyll in plants as a probe to analyze the effects of various stresses on plants. Its use as an effective research method to study the function of photosynthetic mechanisms in plant leaves under adversity [15,16] can rapidly reflect the changes in light energy uptake and utilization capacity of plant leaves under adverse stress [17,18]. The chlorophyll fluorescence technique has the advantages of being fast, sensitive, and non-destructive. It has become an effective method for studying changes in plant physiological status or stress from adversity [19,20]. The chlorophyll fluorescence kinetic OJIP curve contains a large amount of information about the structure and function of photosystem II (PSII), which includes two critical intermediate nodes, the J and I points. Thus, it is possible to divide the plant leaf fluorescence intensity change curve into three main phases: O–J, J–I, and I–P [21]. The O–J phase contains photochemical information, and the J–I and I–P phases include thermal dissipation information. Meanwhile, information about the photosynthetic electron transport chain is provided by many fluorescence parameters derived from OJIP curves.

It has been shown that PWN invasion of pine trees leads to significant changes in variable chlorophyll fluorescence for a brief period after invasion [22]. The net photosynthetic rate of *Pinus thunbergii* and *P. massoniana* infected by the PWN was more significant at the stage of needle discoloration and wilting than at the asymptomatic stage [14]. This experiment investigated the PSII reaction center activity changes and photosynthetic energy partition coefficient of needles under PWN stress by applying the chlorophyll fluorescence induction technique with four-year-old *P. thunbergii* as the experimental material. The research aims were to investigate changes in chlorophyll fluorescence parameters to establish a diagnostic method to evaluate whether pine is susceptible to the disease at the early stage of PWN infection.

## 2. Materials and Methods

### 2.1. Experimental Materials

PWN (AMA3 strains) were provided by the Forest Pathology Laboratory of Nanjing Forestry University. Four-year-old *P. thunbergii* were purchased from Suqian City, Jiangsu Province, China, and planted at Nanjing Forestry University.

### 2.2. PWN Inoculation Experiment and Disease Observation

The experiment was conducted on June 9, 2022. Two groups were set up, and the truncated casing method was used for inoculation [23]. The treatment group (TR) was inoculated with 10,000 AMA3 for 30 plants. At the same time, the control group (CK) was injected with sterile water for a total of 6 plants. Watering and seedling management were conducted every three days. The disease development of *P. thunbergii* was observed, and disease indices were recorded at 1, 4, 7, 10, and 13 days after inoculation.

The calculation of the disease grade and disease index of *P. thunbergii* was based on the method proposed by Tan [24], with slight modifications: grade 0 was normal, with green leaves, and the representative value was 0; grade 1 was for 1/2 or less with chlorotic leaves and 1/4 or less with yellowish leaves, and the representative value was 1; grade 2 was for 1/2 or more with chlorotic leaves and from 1/4 to 3/4 with yellowish leaves, and the representative value was 2; grade 3 was for 3/4 or more with yellowish leaves and 1/2 or less with reddish leaves, and the representative value was 3; and grade 4 was for 1/2 or

more with reddish leaves, and the representative value was 4. The calculation formula was as follows:

$$\text{Disease index} = \Sigma(\text{number of plants of each disease grade} \times 100/(\text{total number of plants} \times \text{representative value of the highest disease grade}). \quad (1)$$

### 2.3. Measurement of Chlorophyll Fluorescence Kinetic Parameters in P. thunbergii Needles

The OJIP parameters of black pine needles were determined using a mini excitation chlorophyll fluorometer (FluorPen FP 110/D, PSI, Brno, Czech Republic). At least five trees were simultaneously measured in the TR and CK groups. Before measuring, the needles needed to be darkened with leaf clips for at least 0.5 hour. Care was taken to fill the measurement holes with needles and avoid any overlap between needles completely.

The OJIP parameters were induced using 90% saturated pulsed light (3000 $\mu mol \cdot m^{-2} \cdot s^{-1}$) at the moments corresponding to 0.051, 2.021, 30.321, and 1001.621 ms for the O, J, I, and P points, respectively. The fluorescence signal was recorded from 11 $\mu s$ until the end of 2 s. A total of 458 data points were recorded. OJIP bar graphs were plotted using the mean values of the OJIP parameters (excluding outliers) for five different measurement times. The raw data were normalized for the O–P and O–J points, and the horizontal coordinates of the parameters were transformed using log Lg as the base using the following formulas:

O-P point normalization:

$$V_{O-P} = (Ft - Fo)/(Fm - Fo) \quad (2)$$

O-J point normalization:

$$V_{O-J} = (Ft - Fo)/(F_J - Fo) \quad (3)$$

note: Ft in the formula is the fluorescence intensity at each time point.

The difference in the OJIP parameters normalized in TR and CK was calculated separately according to the following formulas:

$$\Delta V_{O-P} = V_{O-P}(TR) - V_{O-P}(CK) \quad (4)$$

$$\Delta V_{O-J} = V_{O-J}(TR) - V_{O-J}(CK) \quad (5)$$

the measured OJIP parameters were analyzed using the JIP test to obtain the following variables: chlorophyll fluorescence parameters—the electron transfer status of PSII (Fm/Fo), the maximum photosynthetic quantum yield of PSII (Fv/Fm), the potential activity of PSII (Fv/Fo), the yield of the electron transport per trapped exciton (Psi_o), the quantum yield of electron transport flux (Phi_Eo), the basal quantum yield of the non-photochemical processes in PSII (Phi_Do), the absorption flux per reaction center (ABS/RC), the capture flux per reaction center (TRo/RC), the quantum yield per reaction center (Phi_Do), the absorption flux per reaction center (ABS/RC), the capture flux per reaction center (TRo/RC), and the dissipation flux per reaction center (DIo/RC) for non-photochemical processes in PSII [25,26].

### 2.4. Data Analysis

The mean values of the main chlorophyll fluorescence parameters between the TR and CK groups were calculated using SPSS and analyzed for significant differences. Origin was used to build the graphs. The relative rate of change of each fluorescence parameter was calculated to compare the degree of variation between the chlorophyll fluorescence parameters. The relative rate of change $\Delta$Fm/Fo was calculated using Fm/Fo as an example:

$$\Delta Fm/Fo = [Fm/Fo(TR) - Fm/Fo(CK)]/[Fm/Fo(CK)] \quad (6)$$

## 3. Results

### 3.1. Statistics on the Disease Incidence and Disease Index of P. thunbergii

Thirteen days after inoculation, 10 pines in the TR began to have discolored needles at the inoculation site. Most *P. thunbergii* in TR had discolored needles sixteen days after inoculation. All plants inoculated with PWN died 30 days after inoculation. During the inoculation period, the trees in CK were healthy, and their needles remained green (Table 1).

**Table 1.** Susceptibility and disease index of *P. thunbergii* seedlings after inoculation with PWN.

| Treatment | Disease Incidence/% | | | | | | | Disease Index | | | | | | |
|---|---|---|---|---|---|---|---|---|---|---|---|---|---|---|
| | 1 | 4 | 7 | 10 | 13 | 16 | 30 | 1 | 4 | 7 | 10 | 13 | 16 | 30 |
| AMA3 | 0 | 0 | 0 | 0 | 33.3 | 60 | 100 | 0 | 0 | 0 | 0 | 8.3 | 16.7 | 100 |
| CK | 0 | 0 | 0 | 0 | 0 | 0 | 0 | 0 | 0 | 0 | 0 | 0 | 0 | 0 |

### 3.2. Effect of PWN on the Rapid Chlorophyll Fluorescence Induction Kinetic Curves of P. thunbergii

The OJIP curve contains rich information on the primary photochemical reaction of PSII, which can be studied, including the electron supply and transfer of PSII and the change in PSII reaction center activity. The OJIP curves of *P. thunbergii* needles under PWN stress were altered (Figure 1a). The fluorescence intensities of points O and J in TR were higher than those in CK (Figure 1b). The fluorescence intensity at points O and I increased significantly ($p < 0.05$) after comparing the mean values of fluorescence intensity at O, J, I, and P for the five time points in the TR and CK groups (Figure 1b). The pattern of fluorescence intensity changes at points I and P was not obvious (Figure 1b).

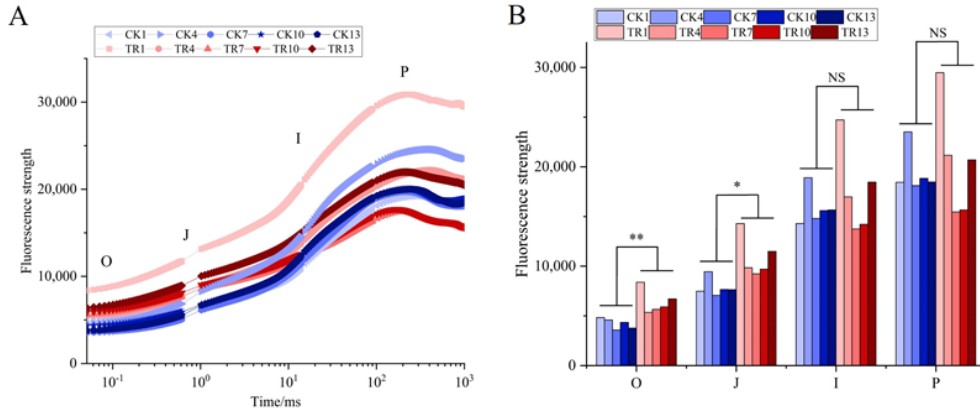

**Figure 1.** Effect of PWN on rapid chlorophyll fluorescence induction kinetic curves of *P. thunbergii* needles. (**A**) The trend of OJIP curves. (**B**) Analysis of the differences between O, J, I, and P points. CK1, CK4, CK7, CK10, and CK13 indicate day 1, day 4, day 7, day 10, and day 13 after inoculation in the control group (CK), whereas TR1, TR4, TR7, TR10, and TR13 indicate day 1, day 4, day 7, day 10, and day 13 after inoculation in the treatment group (TR). * $p < 0.05$, ** $p < 0.01$, and NS: not significant using the Duncan's multiple range test.

### 3.3. Effect of PWN on the Electron Acceptor Side and Electron Donor Side Transfer Capacity of PSII in P. thunbergii

Because the raw OJIP data are susceptible to external factors, the fluorescence data are often standardized by mathematical transformation to facilitate comparisons between data from different treatments. After normalizing the OJIP data, it was found that the relative variable fluorescence intensity at point J was significantly enhanced in the OJIP curve of TR compared with CK. Additionally, the relative variable fluorescence intensity at points J and I was the highest in TR on the 10th day after nematode inoculation. In

contrast, the relative variable fluorescence intensity in CK did not change significantly (Figure 2a,b). It was also shown by the difference of OJIP curves normalized between TR and CK at different times that the relative variable fluorescence intensity of *P. thunbergii* changed significantly at point J after inoculation with PWN. In addition, the relative variable fluorescence intensity at point J reached its maximum on the 10th day after inoculation. Point I had a trend of enhancement several times except for the decrease of relative variable fluorescence intensity on the 4th day after inoculation with nematodes (Figure 2c, d). The fluorescence intensity of point K on the OJIP curve of *P. thunbergii* needles under PWN stress increased at 0.3 ms. It reached its maximum variable fluorescence intensity on the seventh day after inoculation.

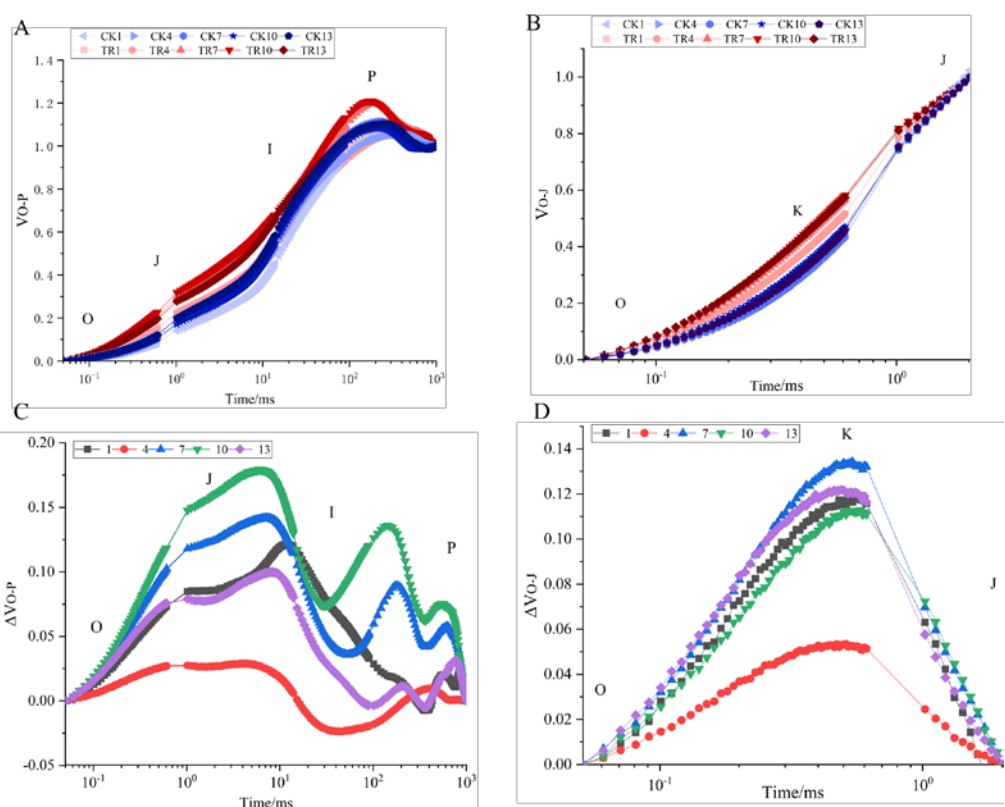

**Figure 2.** Analysis of O-P and O-J curves to standardize *P. thunbergii* needles under PWN stress. (**A**) Trends of O-P normalized fluorescence curves; (**B**) Trends of O-J normalized fluorescence curves. (**C**) Trends of O-P standardized difference fluorescence curves; (**D**) Trends of O-J standardized difference fluorescence curves. The numbers 1, 4, 7, 10, and 13 indicate days 1, 4, 7, 10, and 13 after inoculation.

### 3.4. Effect of PWN on Chlorophyll Fluorescence Parameters of P. thunbergii Needles

The light energy absorption and distribution parameters of *P. thunbergii* needles under PWN stress changed significantly (Figure 3). Compared with CK, Fm/Fo, Fv/Fm, and Fv/Fo in TR showed a significant decreasing trend, indicating that the light energy conversion efficiency and potential viability in PSII decreased after the invasion of PWN into *P. thunbergii*. Psi_o and Phi_Eo in TR also had a decreasing trend, indicating that the openness of the active reaction centers and the quantum yield of electron transfer flux decreased. Phi_Do in TR had an increasing trend, indicating that the quantum yield of the non-photochemical burst increased. ABS/RC in TR had a more apparent increasing trend. This indicated that the decrease in the number of active reaction centers in the PSII reaction centers forced the remaining active reaction centers to increase their efficiency. An increase in light energy absorption was shown. TRo/RC in TR had an increasing trend, indicating that the number of active reaction centers in the PSII reaction centers decreased.

The increase in TRo/RC and DIo/RC in TR meant that the proportion of light energy absorbed by PSII reaction centers for photosynthetic electron transfer energy and heat dissipation increased, and the increase in QA- reduction captured more energy per reaction center.

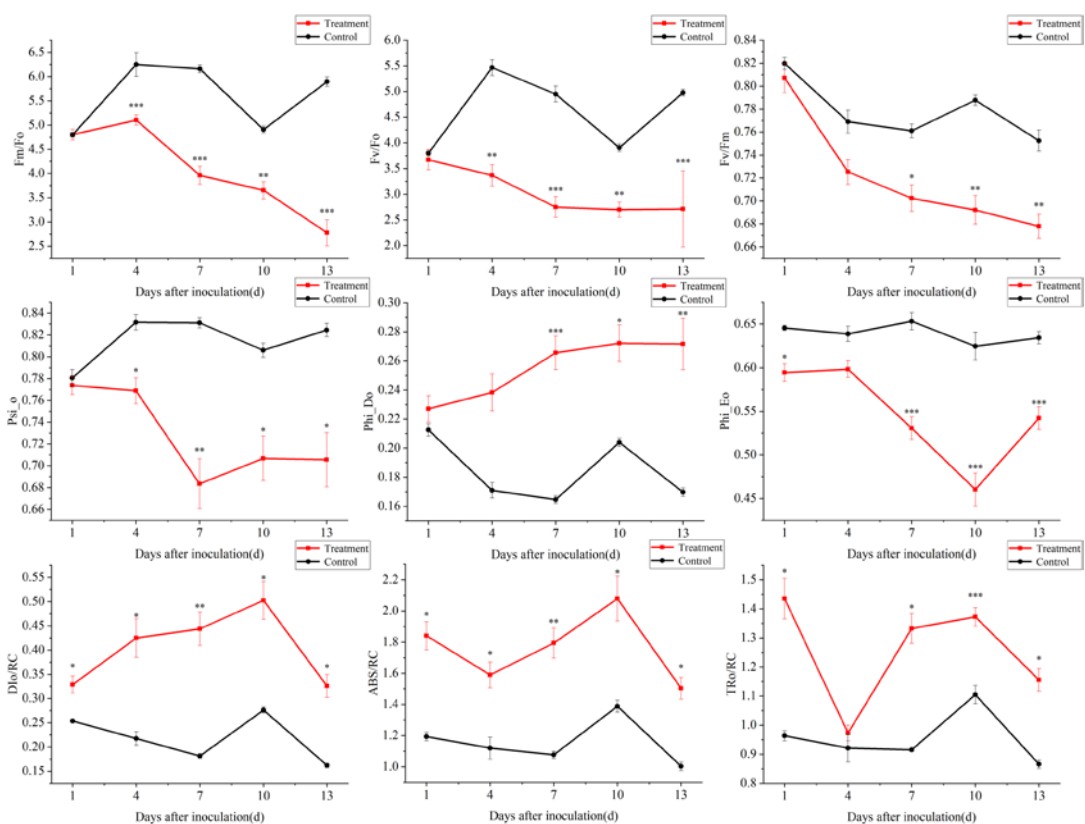

**Figure 3.** Analysis of chlorophyll fluorescence parameters of *P. thunbergii* needles under PWN stress. * $p < 0.05$, ** $p < 0.01$, *** $p < 0.001$ using Duncan's multiple range test.

Among the relative change rates of chlorophyll fluorescence parameters (Table 2), the change in degree of DIo/RC, ABS/RC, and TRo/RC was more prominent on day one after inoculation. The degree of change in ABS/RC was the largest, and Fm/Fo was the smallest. On day four after inoculation, the degree of change in Fv/Fo, Phi_Do, DIo/RC, and ABS/RC was more prominent. On the seventh day after inoculation, Fm/Fo, Fv/Fo, Phi_Do, DIo/RC, ABS/RC, and TRo/RC changed more prominently; DIo/RC changed the most, and Psi_o changed the least. On the 10th day after inoculation, Fv/Fo, Phi_Do, DIo/RC, and ABS/RC changed to a more prominent extent, DIo/RC changed to the greatest extent, and Psi_o changed to the least extent. On the 13th day after inoculation, the *P. thunbergii* began to show symptoms; at this time, Fm/Fo, Fv/Fo, Phi_Do, DIo/RC, ABS/RC, and TRo/RC changed to a more prominent extent, DIo/RC changed to the greatest extent, and Psi_o changed to the least extent. It was shown that these chlorophyll fluorescence parameters, Fv/Fo, Phi_Do, DIo/RC, and ABS/RC, were better indicators for early diagnosis, with the best combined performance with DIo/RC.

**Table 2.** The relative rates of change in chlorophyll fluorescence parameters of *P. thunbergii* after inoculation with PWN.

| Days after Inoculation /% | The Relative Rates of Change in Chlorophyll Fluorescence Parameters | | | | | | | | |
|---|---|---|---|---|---|---|---|---|
| | ΔFm/Fo /% | ΔFv/Fo /% | ΔFv/Fm /% | ΔPsi_o /% | ΔPhi_Do /% | ΔPhi_Eo /% | ΔDIo/RC /% | ΔABS/RC /% | ΔTRo/RC /% |
| 1 | 0.107 | −3.300 | −0.875 | −1.539 | 6.866 | −7.916 | 29.728 | 54.155 | 48.888 |
| 4 | −18.340 | −38.398 | −7.535 | −5.689 | 39.374 | −6.351 | 95.139 | 41.942 | 5.620 |

| 7 | −35.692 | −44.484 | −17.756 | −7.699 | 61.330 | −18.749 | 144.725 | 66.697 | 45.578 |
| 10 | −25.521 | −30.927 | −12.316 | −12.135 | 33.402 | −26.301 | 81.783 | 49.739 | 24.224 |
| 13 | −52.893 | −45.520 | −14.428 | −9.918 | 59.959 | −14.529 | 101.160 | 49.950 | 33.453 |

## 4. Discussion

It was revealed by the results of the study that the K point rose in the JIP-test analysis of *P. thunbergii* under PWN stress. The rise in K points due to salt stress was also observed in maize [27] because it disrupts the oxygen-emitting complex (OEC) and impairs the electron transfer capacity of the PSII donor side. Since OEC is involved in the photo-oxidation of water during the light reaction of photosynthesis, it is hypothesized that the oxygen-evolving complex is impaired in *P. thunbergii* needles under PWN stress. Fv/Fm, Psi_o, and Phi_Eo had a decreasing trend, indicating a decrease in the opening of active reaction centers and a decrease in the quantum yield of electron transfer flux. Researchers have obtained equivalent results in studies of other plant diseases [28–30]. Damage to the OEC on the electron donor side of PSII was indicated in all studies [31].

ABS/RC and TRo/RC appeared to rise at an early stage of PWN infection in *P. thunbergii*. The rise in ABS/RC and TRo/RC in transgenic mustard under heavy metal stress implies the inactivation of certain RCs [32]. Thus, PWN would likely lead to a decrease in the number of active reaction centers in the PSII reaction centers of *P. thunbergii* needles. The increase in DIo/RC indicates the excess excitation energy used for heat dissipation [32]. This may be a resistance mechanism generated by *P. thunbergii* to PWN stress. An increase in the proportion of photosynthetic electron transfer energy would inevitably increase the generation of assimilative power and, thus, carbon assimilation capacity [33] and produce more defense substances. Therefore, under PWN stress, there is a change in the ability of the photosynthetic apparatus of *P. thunbergii* needles to absorb and utilize light energy. Additionally, there is an increase in the proportion of energy absorbed by the PSII reaction centers of *P. thunbergii* needles for electron transfer after QA- and the energy captured per reaction center for electron transfer. This is one of the essential manifestations of the positive response of the photosynthetic apparatus of *P. thunbergii* to resist its damage. Additionally, higher plants protect PSII by initiating a nonradiative energy dissipation mechanism dependent on the xanthophyll cycle and a reduction in the activity of reaction centers [34]. Early in disease development, *P. thunbergii* may reduce the degree of photoinhibition of needle PSII under PWN stress by initiating the xanthophyll cycle to combat the massive inactivation of PSII reaction centers.

Invasion by PWN leads to a large amount of $H_2O_2$ being produced by *P. thunbergii* in a short time [35], and chloroplasts are a significant site for the intracellular production of ROS [36] and play an essential role in the defense response [37]. It has been shown in many studies that when plants are subjected to adversity stresses, such as drought and pathogenic bacteria, leaf cells rapidly produce substantial amounts of substances, such as ROS, for defense responses [38,39]. However, ROS can impact photosynthesis-related structures and functions [40]. Therefore, it may not be possible to distinguish whether pine trees are affected by PWN or drought using chlorophyll fluorescence parameters.

The chloroplast itself does not contain a catalase (CAT) that scavenges $H_2O_2$, and excess $H_2O_2$ impairs the function of PSII [41]. When the activity of OEC is inhibited, it intensifies the production of $H_2O_2$ during hydrolysis [42]. Therefore, PWN invasion may reduce photosynthetic electron production by inhibiting the OEC activity of *P. thunbergii* needles. The OEC inhibition of OEC activity will intensify ROS production in *P. thunbergii*, thus disrupting intracellular metabolism [43], eventually leading to cell necrosis. In addition, PWN stress caused a decrease in the electron transfer rate and disrupted light energy utilization in *P. thunbergii* needles, which is the leading cause of photoinhibition.

Although the traditional chlorophyll fluorescence technique is expected to achieve an early diagnosis of PWD at the leaf level, it cannot address the spatial variation in response to the same leaf. The sample trees selected for the experiment were young enough to provide a reference for future field studies on large trees. Still, factors such as canopy

position and season of adult pine trees in the field may impact leaf chlorophyll fluorescence information [44,45], and further validation of the method is needed. Additionally, whether other pine pests, diseases, and different regions affect the discrimination of the results needs to be further investigated.

## 5. Conclusions

In this study, it was explored how PWN stress affected the PSII function of the photosynthetic apparatus in *P. thunbergii* seedlings using chlorophyll fluorescence kinetics. Chlorophyll fluorescence curves and related parameters changed significantly after *P. thunbergii* was subjected to PWN stress. These parameters are expected to be used as indicators for early PWD diagnosis.

**Author Contributions:** Conceptualization, experimental, data analysis, manuscript—writing, F.L.; writing—article guidance and data analysis, M.-J.Z.; sample collection, J.-F.H., M.P. and L.-Y.S.; guarantor of the integrity of the entire study and approval of the final version of the manuscript, J.-J.T. and J.-R.Y. All authors have read and agreed to the published version of the manuscript.

**Funding:** This research was funded by the National Key Research and Development Program of China (2021YFD1400900) and the Major Emergency Science and Technology Project of the National Forestry and Grassland Administration (ZD202001).

**Informed Consent Statement:** Informed consent was obtained from all subjects involved in the study.

**Data Availability Statement:** The data presented in this study are available on request from the corresponding author.

**Conflicts of Interest:** The authors declare no conflict of interest.

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
