# Peer review of "Early Diagnosis of Pine Wilt Disease in Pinus thunbergii Based on Chlorophyll Fluorescence Parameters"

_forests, doi:10.3390/f14010154_

Round 1

Reviewer 1 Report

Introduction is well prepared, aims of your research are described however it will be betted to point out them more prominently at the end of this part e.g. in the form “Research aims were i) to establish…ii) to evaluate …etc”.

Table 2: please think about including values from control variants in the table.

Discussion part is good and quite detailed, but I would prefer to add more remarks on possibilities of distinguishing between biotic and abiotic stress using your measurements, will it be possible to discriminate stress inducted to pine trees by drought and PWN? Simple remark on this could be also incorporated into Conclusion.

Author Response

1: Introduction is well prepared, aims of your research are described however it will be betted to point out them more prominently at the end of this part e.g. in the form “Research aims were i) to establish…ii) to evaluate …etc”.

Response 1: Thank you for your suggestion. I have revised the introduction. The revised details can be found in L75-80.

2: Table 2: please think about including values from control variants in the table.

Response 2: Thank you for your suggestion. Table 2 is calculated from the data in Figure 3 using the L135 formula, whether the control group variables still need to include.

3: Discussion part is good and quite detailed, but I would prefer to add more remarks on possibilities of distinguishing between biotic and abiotic stress using your measurements, will it be possible to discriminate stress inducted to pine trees by drought and PWN? Simple remark on this could be also incorporated into Conclusion.

Response 3: Your suggestion is very good. I have revised the discussion. The revised details can be found in L259-262.

Reviewer 2 Report

Original and relevant topic to the field, exploring new avenues for rapid diagnostics and early detection of the pinewood nematode, one of the most notorious threats to conifers worldwide. This opens up new possibilities for a much-needed rapid detection of the nematode’s presence in susceptible pine trees. Giving some suggestions below to improve the manuscript.

Major comments:

Abstract & Introduction: well-structured and references are appropriate.

Material & Methods: experimental design is good, experiments are reproducible and methods are adequate. Proper references to previously published methodology. Sound and comprehensive statistical analysis.

Results & Discussion: results, including tables and figures, are clearly labelled and nicely presented. Logical interpretation of results and conclusions.

Specific comments:

Introduction

Line 35: reference 7 is outdated (2009) and claiming that the “PWN is a harmful invasive organism in 52 countries” is incorrect. That would imply that pine wilt disease would potentially be reported in those 52 countries, and it’s not the case... It may have been intercepted in many countries, but thankfully the nematode did not establish in all of them. Refer to this link for accurate and current distribution data and adapt/change the text accordingly: https://gd.eppo.int/taxon/BURSXY/distribution

Lines 36-37: what do authors mean by “The pathogenic mechanism of PWN has not been clarified”? On a molecular level, perhaps, but the pathogenicity of the PWN is extensively documented. What makes managing PWD challenging is that it results from complex interactions between the nematode, a susceptible plant host and the insect vector.

Line 52: decrease rather than “decreased”

Line 70: Pinus thunbergii and P. massoniana

Line 70: infected by the PWN; PWD is a consequence of the presence of the nematode

Line 71: “[...] was more significant at the stage of needle discoloration and wilting than at the stage of health.” – I would rephrase to “[...]and wilting than at the asymptomatic stage.”

Results

Line 139: “All plants inoculated with PWN died 30 after inoculation” – 30 what? I’m assuming days, but the word is missing.

The inoculum used is quite high for 4-year-old plants, so the changes in physiological parameters was expected, especially considering young plants are more susceptible to pathogens that older ones. Out of curiosity, would authors expect similar results in terms of early detection if they inoculated 1000 nematodes per plant?

Author Response

1: Line 35: reference 7 is outdated (2009) and claiming that the “PWN is a harmful invasive organism in 52 countries” is incorrect. That would imply that pine wilt disease would potentially be reported in those 52 countries, and it’s not the case... It may have been intercepted in many countries, but thankfully the nematode did not establish in all of them. Refer to this link for accurate and current distribution data and adapt/change the text accordingly: https://gd.eppo.int/taxon/BURSXY/distribution.

Response 1: Your suggestion is very good. We are sorry that we misunderstood you due to an error on our part. We intended to say that PWN is listed as a harmful invasive organism in 52 countries and that the most recent documented reference was found to be reference 7. This document is really out of date, so we have removed this sentence from this article. The revised details can be found in L35.

2: Lines 36-37: what do authors mean by “The pathogenic mechanism of PWN has not been clarified”? On a molecular level, perhaps, but the pathogenicity of the PWN is extensively documented. What makes managing PWD challenging is that it results from complex interactions between the nematode, a susceptible plant host and the insect vector.

Response 2: Thank you for your suggestion. About pathogenetic mechanism of PWN there are some major opinions. First, the enzymes especially cellulase reproduced by PWN destroyed the cell wall and cell membrane of parenchymatous cells. The oleoresin leaked and diffused into tracheids unnormally caused water deficiency and wilt of pines. Second, after infection of pines with PWN volatile terpenes increased in xylem tissue. The terpenes came into tracheids caused cavitation in the tracheids and the cavitation interrupted water-conduction in sapwood. Third, toxins were produced by pines or bacteria after infection of pines by PWN. These toxins could arouse pine wilt. So, we think the pathogenic mechanism of PWN has not been clarified.

3: Line 52: decrease rather than “decreased”.

Response 3: Thank you for your suggestion. The revised details can be found in L52.

4: Line 70: Pinus thunbergii and P. massoniana.

Response 4: Thank you for your suggestion. The revised details can be found in L70.

5: Line 70: infected by the PWN; PWD is a consequence of the presence of the nematode.

Response 5: Thank you for your suggestion. The revised details can be found in L70.

6: Line 71: “[...] was more significant at the stage of needle discoloration and wilting than at the stage of health.” – I would rephrase to “[...]and wilting than at the asymptomatic stage.”

Response 6: Thank you for your suggestion. The revised details can be found in L71-72.

7: Line 139: “All plants inoculated with PWN died 30 after inoculation” – 30 what? I’m assuming days, but the word is missing.

Response 7: Thank you for your suggestion. The revised details can be found in L141.

8: The inoculum used is quite high for 4-year-old plants, so the changes in physiological parameters was expected, especially considering young plants are more susceptible to pathogens that older ones. Out of curiosity, would authors expect similar results in terms of early detection if they inoculated 1000 nematodes per plant?

Response 8: Thank you for your suggestion. For Pinus thunbergii, both seedlings and adult trees are susceptible. We speculate that similar results will occur in terms of early detection if they inoculated 1000 nematodes per plant, but the onset time of P. thunbergii will be delayed, which will have a certain impact on the results, resulting in delayed changes in chlorophyll fluorescence parameters.

Reviewer 3 Report

The manuscript describes the development of a tool in basic and applied research on plant physiology and agronomy, for early diagnosis of pine wilt disease in Pinus thunbergii based on chlorophyll fluorescence parameters.

In general, the manuscript is interesting and well-written. In the results section,  the writing can be improved. 

I have a few comments for the authors. I recommend a minor revision of the manuscript.

Please find my comments on your paper in the attached document.

Author Response

1: Please, give the abbreviation"Hydrogen peroxide (H2O2)".

Response 1: Thank you for your suggestion. The revised details can be found in L44.

2: Please, add the abbr. with full name For example "photosystem II (PSII)".

Response 2: Thank you for your suggestion. The revised details can be found in L62.

3: Is there any specific reason for choosing these days? If so, please add a reference.

Response 3: Thank you for your suggestion. We found that Pinus thunbergii started to show signs around 15 days when inoculated with 10,000 nematodes per plant in our previous pre-experiments, so this experiment was set up to measure the data at such a time.

3: Please avoid using the same phrases in sentences.

Response 4: Thank you for your suggestion. The revised details can be found in L211-213.